# Investigating the Calcination Temperature and Grinding Time of Calcined Clay on the Mechanical Properties and Durability of LC3 Concrete

Sina Nasiri *, Rahmat Madandoust and Malek Mohammad Ranjbar

Department of Civil Engineering, University of Guilan, Rasht 41996-13776, Iran;
rmadandoust@guilan.ac.ir (R.M.); ranjbar@guilan.ac.ir (M.M.R.)
* Correspondence: sinanasiri@phd.guilan.ac.ir

**Abstract:** The impact of the calcination temperature and grinding time on the mechanical properties and durability of limestone-calcined clay concrete (LC3) is crucial. In this research, calcined clay was produced within the temperature range of 700 °C to 900 °C, and the grinding time varied from 15 to 120 min. This study examines compressive strength and chloride penetration resistance using the rapid chloride migration test on LC3 concrete over a period of 180 days. The findings reveal that clay calcined at 800 °C for 120 min exhibited the highest specific surface area compared to other calcined clays. Furthermore, the compressive strength of LC3 concrete incorporating clay calcined at 800 °C for 60 min surpassed that of other mixtures investigated. Additionally, the chloride diffusion coefficient of LC3 concrete with calcined clay prepared at 800 °C for 120 min was lower than other mixtures.

**Keywords:** grinding time; calcination temperature; calcined clay concrete; chloride penetration





## 1. Introduction

The cement industry, contributing to approximately 7.6% of $CO_2$ emissions, is among the major industries with significant environmental impacts. A reduction in $CO_2$ emissions from this industry can play a crucial role in mitigating climate change hazards [1,2]. Supplementary cementitious materials (SCMs) have gained significant attention as partial replacements for conventional Portland cement, contributing to reduced $CO_2$ emissions and the conservation of valuable minerals in cement production [2–5]. Among the emerging SCMs, the combination of calcined clay and limestone, known as calcined clay cement (LC3), has drawn global interest due to the abundant availability of these materials worldwide [6]. Calcined clay possesses a high content of the amorphous reactive phase, making it a reactive SCM with great potential. LC3 offers promising advantages, including energy and $CO_2$ emissions reduction compared to conventional Portland cement production [4,7,8].

Various studies have reported on the mechanical properties of LC3 [9–13]. Notably, the synergistic effects of limestone and calcined clay have shown positive impacts on the early strength of mortar [9]. Furthermore, in recent years, the advantages of using various supplementary cementitious materials along with cement in concrete, known as ternary concrete, have been demonstrated [14–17]. Based on this, LC3 concrete can also be classified as ternary concrete.

In a study by Marangu [18], the compressive strength of LC3 concrete was compared with ordinary Portland cement concrete and pozzolanic-blended cement concrete at three different water-to-cementitious material ratios. The compressive strength of ordinary Portland cement concrete was higher than that of LC3 and pozzolanic-blended cement concrete at ages between 2 and 28 days. However, at 90 and 180 days, the compressive strength of LC3 concrete exceeded that of ordinary Portland cement concrete and pozzolanic-blended concrete.

In the research conducted by Maraghechi et al. [3], the influence of calcined clay containing varying percentages of kaolinite, along with stone powder in the form of LC3, as well as a separate cementitious supplement, on the compressive strength of mortar has been investigated. The compressive strength of LC3 mortars with clay soil containing over 40% kaolinite and 50% cement replacement with calcined clay soil showed similar or higher strength than mortars with Type 1 Portland cement from day 7 onwards.

Contrary to the aforementioned results, in the study by Lin et al. [19], the replacement of 30% cement with calcined clay and stone powder resulted in a decrease in the compressive strength of LC3 mortar at 180 days compared to the control mortar. Additionally, in the research conducted by Nguyen et al. [20], the impact of replacing 15% and 20% of cement with calcined clay along with stone powder on compressive strength was investigated. The findings of this study indicate that at a replacement rate of 15%, the compressive strength of LC3 concrete was higher than the control concrete from 14 days onwards. However, at a replacement rate of 20%, LC3 concrete exhibited lower compressive strength than the control concrete for up to 28 days.

The results of the research by Bahman-Zadeh et al. [21] suggest that the application of calcined clay, either in the form of LC3 or as a separate cementitious additive, led to a reduction in strength for up to 90 days compared to the controlled mixture. Furthermore, these findings indicate that the use of calcined clay in the form of LC3 shows better performance compared to using it solely as a cementitious additive.

There exist discrepancies in the trend of LC3 compressive strength results in comparison to Portland cement concrete across different investigations. Some studies report comparable or higher compressive strengths for LC3 concrete compared to ordinary Portland cement concrete [3,13,22], while others indicate lower resistance, even after 360 days of curing [19,20,23,24].

The durability properties of LC3-based mortars and concretes in chloride environments have also been extensively studied [3,5,25–27]. Generally, LC3 has been found to enhance concrete's resistance to chloride penetration compared to conventional Portland cement concrete. Research by Dhandapani et al. [12] showed significantly lower chloride ion penetration in LC3 binders compared to Portland cement binders when evaluated using the RCMT method. Similarly, Nguyen et al. [20] found that the addition of 15% and 20% of calcined clay and limestone reduced the diffusion coefficient (RCMT) by approximately 40% and 55%, respectively, compared to OPC concrete.

However, variations in the mechanical and durability properties of LC3 can largely be attributed to the pozzolanic performance of calcined clay, which is influenced by its manufacturing process. Although many studies have focused on the impact of soil types and calcination temperature in binary cement mixture properties [28–31]. As an example, the study conducted by Fernandez et al. [31] can be referred to, in which the impact of three types of calcined clay, namely montmorillonite, kaolinite, and illite, at two temperatures of 600 and 800 °C, during their pozzolanic performance in cementitious blends has been investigated. Kaolinite was shown to have the highest potential for activation. The pozzolanic activity of calcined kaolinite can enhance the mechanical properties of cement blends, surpassing that of the 100% cement reference paste, which was already on day 7.

Some studies have investigated the effects of the calcination process (calcination temperature and fineness) on the pozzolanic properties of calcined clay in cementitious matrices [31–33]. Balykov's research, for instance, investigated the effect of calcination at temperatures ranging from 500 to 900 °C, identifying that calcined clay at 700 °C for 2 h exhibited the best performance in the binary cement matrix [33].

In the study by Ferreiro et al. [32], the effect of calcination temperatures at 700, 850, and 1000 °C on the performance of calcined clay in LC3 blends was examined. The optimal calcination temperature reported in this research was 850 °C. In this research, the furnace temperature increased instantly to the desired temperature, while the performance of calcined clay in the mixture decreased. Moreover, the investigation of compressive strength results indicated that, for up to 90 days, the compressive strength of mixtures containing

calcined clay and stone powder was lower than that of Portland cement mixtures. However, at 90 days, the difference in compressive strength between LC3 mixtures and control mixtures was negligible.

Furthermore, the method of obtaining supplementary cementitious materials, such as rice husk ash and calcined clay, can also impact their quality and performance in cement mixtures. Controlled burning involves gradually heating raw material to the desired temperature in the furnace, followed by cooling and grinding [14,15], whereas instantaneous burning places the material immediately at the specified temperature [21,31–33]. The combustion method can affect the quality of the SCMs, as observed in the case of rice husk ash. Rice husk ash is mentioned because, much like calcined clay, it undergoes a combustion process during its preparation. The research findings have demonstrated that an abrupt increase in furnace temperature can to the desired level and result in a reduction in the specific surface area of rice husk ash with a darker coloration. These changes indicate diminished pozzolanic activity in the obtained rice husk ash under such conditions [34–36].

However, the effect of the combustion method on the quality of calcined clay and its performance in concrete remains unexplored. Therefore, this study investigates the influence of the combustion method on the performance of calcined clay. Additionally, this study examines the effects of calcination temperature and grinding time for calcined clay on the compressive strength and chloride diffusion resistance of concrete when measured over a period of up to 180 days.

## 2. Experimental Program

### 2.1. Materials Used and Mixture Properties

The cement utilized in this study was Type II Portland cement produced by the Tehran cement factory. The chemical composition properties of Portland cement, raw clay, and limestone powder (provided by PARS Company, Qom, Iran) is presented in Table 1. The compound composition of the cement included 52.6% C3S, 22.8% C2S, 4.8% C3A, and 10.9% C4AF. In Table 2, a comparison of the physical properties of cement with ASTM C150 [37] requirements are presented. The raw clay was provided by the Iran China Clay Industries Company (ICC KAOLIN). The clay used was kaolinite-type clay.

**Table 1.** Chemical composition of cement, limestone, and raw clay.

| Composition % | $SiO_2$ | $Al_2O_3$ | $Fe_2O_3$ | CaO | MgO | $SO_3$ | $Na_2O$ | $K_2O$ | L.O.I. | IR |
|---|---|---|---|---|---|---|---|---|---|---|
| Cement | 21.8 | 4.1 | 3.58 | 62.7 | 1.9 | 1.51 | 1.2 | 1.46 | 1.61 | 0.75 |
| Limestone (LS) | 3.28 | 0.75 | 0.22 | 49.3 | 2.82 | 0.09 | - | 0.03 | 41.6 | - |
| Raw clay | 60.61 | 29.72 | 1.38 | 1.08 | 0.41 | - | 0.09 | 0.34 | 9.31 | - |

**Table 2.** Comparison of physical properties of cement with ASTM C150 requirements.

| Properties | Cement | ASTM C150 Requirements |
|---|---|---|
| Specific surface ($cm^2$/gr)-min | 3060 | 2600 |
| Time of setting (minuts) | | |
| Initial min | 190 | 45 |
| Final-max | 235 | 375 |
| Compressive strength (MPa) | | |
| 3 days-min | 14.1 | 10.0 |
| 7 days-min | 23.5 | 17.0 |

The calcination of the clay was carried out at temperatures of 700 °C, 800 °C, and 900 °C for a duration of 1 h. The temperature was gradually raised from ambient to the desired temperature at a heating rate of 10 °C/min. Figure 1 provides a visual representation of the time–temperature variations during the controlled combustion process. Subsequently,

calcined clay was immediately removed from the furnace and rapidly cooled to room temperature. It is important to note that the cooling rate was not considered in this study. After the calcination process, calcined clay was subjected to milling at four different time intervals: 15, 30, 60, and 120 min. For instantaneous combustion (where temperature was immediately raised from ambient to a specific temperature), calcined clay was produced at a temperature of 800 °C and a grinding time of 60 min.

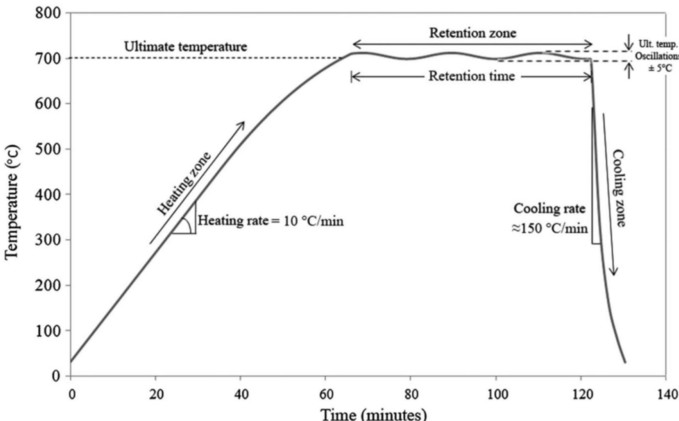

**Figure 1.** Typical time–temperature variations for controlled combustion.

Drinking water was used to prepare concrete blends along with a polycarboxylic ether-type superplasticizer. The aggregates utilized in these mixtures included river sand with a density of 2510 kg/m$^3$ and crushed coarse aggregate with a maximum size of 19 mm and a density of 2590 kg/m$^3$, meeting the requirements of ASTM C 33 [38].

Concrete mixtures were produced with a water–binder ratio of 0.4 and a total amount of cementitious materials at 400 kg/m$^3$. The selected water–binder ratio was 0.4 for all mixes. The mixing proportions of the control mix are detailed in Table 3. For the LC3 mixtures, 25% of Portland cement was replaced with calcined clay, and an additional 5% was substituted with limestone powder.

**Table 3.** Mixture proportions of the control mixture.

| Mix Designation | W/C | Cement (kg/m$^3$) | LS (kg/m$^3$) | Calcined clay (kg/m$^3$) | Water (kg/m$^3$) | SP (% cement) | Coarse agg. SSD (kg/m$^3$) | Fine agg. SSD (kg/m$^3$) |
|---|---|---|---|---|---|---|---|---|
| Ref | 0.40 | 400.0 | - | - | 160.0 | 0.47 | 867 | 867 |

To maintain consistent workability, the slump value of all concrete mixes was controlled within a range of 115 ± 35 mm by adjusting the dosage of the polycarboxylate ether-type superplasticizer.

*2.2. Tests Performed*

In order to determine the effect of preparation methods on the physical properties of calcined clay, experiments including specific gravity, residue on a 45-micron sieve, a specific surface area using the Blaine method, and nitrogen adsorption were conducted. Given the classification of calcined clay within the group of natural pozzolans, its pozzolanic activity was determined as per ASTM C311 [39], specifying a replacement rate of 20%. However, due to the high purity percentage of $SiO_2$ in calcined clay, its pozzolanic activity was also assessed according to ASTM C1240 [40] with a 10% cement replacement using the accelerated method. Additionally, chemical analysis and XRD tests were performed on calcined clays.

The pozzolanic activity index of the calcined clay was determined using the ASTM C311 [39] method, where 20% of OPC (ordinary Portland cement) was replaced with

calcined clay. Additionally, due to the highly specific surface area of calcined clay, the pozzolanic index 'h' was determined by the ASTM C1240 [40] method, with a 10% replacement of OPC with calcined clay.

The specific surface area of the calcined clay was determined using the nitrogen adsorption instrument, model BELSORP—mini II, and the Blaine apparatus, model cenco no. 15413 from the central scientific company.

Compressive strength testing was conducted at 3, 7, 28, 90, and 180 days on 100 mm cubic specimens following the BS EN 12390 Part 3 [41] standard. Three specimens were tested for each time period.

The chloride rapid test was performed according to the NTBuild 492 [42] method. In this test, cylindrical samples with a diameter of 10 cm and a thickness of 5 cm were exposed to a 10% sodium chloride solution on one side and a 0.3 N sodium hydroxide solution on the other side. The penetration of chloride ions into the samples was accelerated by applying a suitable electrical potential, which was determined based on the initial current flow through the sample. The test setup is depicted in Figure 2.

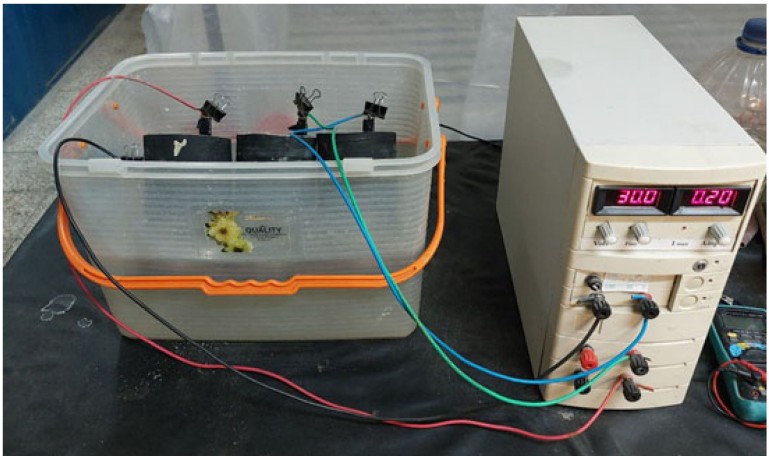

**Figure 2.** Test setup of the rapid chloride migration test.

The test was conducted for a predetermined time, depending on the applied potential. After completing the test, the samples were divided, and a 0.1 N solution of silver nitrate was sprayed onto the divided surfaces. The reaction of silver nitrate with chlorides resulted in the formation of white-colored silver chloride, indicating the chloride penetration front. The migration coefficient was then determined using Equation (1) [42]:

$$D_{nssm} = \frac{0.0239(273 + T)L}{(U - 2)\,t} \left( x_d - 0.0238\sqrt{\frac{(273 + T)\,L\,x_d}{U - 2}} \right) \tag{1}$$

where

$D_{nssm}$ = Non-steady-state diffusion coefficient ($\times\,10^{-12}$ m$^2$/s);
$U$ = Absolute value of the applied voltage (V);
$T$ = Average value of the initial and final temperatures in the anolyte solution (°C);
$L$ = Thickness of the specimen (mm);
$x_d$ = Average value of the penetration depths (mm);
$t$ = Test duration (hours).

## 3. Results and Discussion

### 3.1. Properties of Calcinated Clay

In Table 4, the results of the experiments for determining specific gravity, residue on a 45-micron sieve, specific surface area using the Blaine and nitrogen adsorption methods, and the L.O.I. of the calcined clays and stone powder are presented. According to the

requirements of ASTM C618 [43], natural pozzolans must have a maximum residue on a 45-micron sieve, and their pozzolanic activity indices should be less than 34% and greater than 75% at 7 and 28 days, respectively. According to the numbers in Table 4, calcined clay meets these ASTM standards.

**Table 4.** Physical properties of calcined clays, and limestone.

| | Grinding Time (min) | Specific Surface Area | | Specific Gravity(g/cm$^3$) | Percent Retained on 45 μm Sieve | L.O.I. (%) |
|---|---|---|---|---|---|---|
| | | BET (m$^2$/g) | Blaine (cm$^2$/g) | | | |
| LS | - | - | 4640 | 2.61 | 11.1 | 41.6 |
| 700 °C/1 H/SL | 15 | 20.8 | 4130 | 2.57 | 15.6 | 0.94 |
| | 30 | 21.1 | 4660 | 2.57 | 14.3 | |
| | 60 | 21.3 | 4970 | 2.57 | 12.3 | |
| | 120 | 21.4 | 5090 | 2.57 | 12.3 | |
| 800 °C/1 H/SL | 15 | 35.2 | 5020 | 2.57 | 15.6 | 0.83 |
| | 30 | 35.4 | 5560 | 2.57 | 14.3 | |
| | 60 | 35.5 | 5840 | 2.57 | 12.3 | |
| | 120 | 35.6 | 5980 | 2.57 | 12.3 | |
| 900 °C/1 H/SL | 15 | 30.7 | 4760 | 2.57 | 15.6 | 0.64 |
| | 30 | 30.9 | 5250 | 2.57 | 14.3 | |
| | 60 | 31.0 | 5530 | 2.57 | 12.3 | |
| | 120 | 31.0 | 5670 | 2.57 | 12.3 | |
| 800 °C/2 H/SL | 60 | 23.8 | 5030 | 2.57 | 14.6 | 0.78 |
| 800 °C/1 H/Su | 60 | 19.2 | 4600 | 2.57 | 17.2 | 0.91 |

The investigation of specific gravity and residue on a 45-micron sieve indicates that variations in calcination temperature and the use of the instantaneous temperature increase method did not have a significant impact on them. Increasing the calcination temperature from 700 to 800 °C initially resulted in an increase in the specific surface area of the calcined clay. However, with further temperature increases to 900 °C, the specific surface area decreased. The specific surface area increased by approximately 18% and 67% using the Blaine method and nitrogen adsorption method, respectively, with a temperature increase from 700 to 800 °C. With an increase to 900 °C, the specific surface area had a reduction of around 5% and 13% compared to the temperature of 800 °C for the respective methods. The specific surface area determined using the Blaine method is less sensitive to changes in calcination temperature compared to the nitrogen adsorption method. With an increase in temperature, initial components of clay, such as water, are lost. Then, at temperatures above approximately 450 °C, kaolinite transforms into metakaolin. With further temperature increases, at around 900 °C and beyond, silica transforms into quartz and silicon spinel. The reduction in specific surface area at 900 °C compared to 800 °C could be attributed to alterations in the arrangement of aluminum (AL) structures and changes in the number of quartz crystals and other crystalline phases [31,32,44–46].

The trend of changes in the specific surface area of this research is similar to what was reported by Ferreiro et al. [32]. The specific surface area of calcined clay prepared in the instantaneous temperature increase method has experienced a reduction of approximately 21% and 46% compared with the specific surface area of the calcined clay prepared in the controlled method at 800 °C, using the Blaine and nitrogen adsorption methods, respectively. Therefore, the reason for significant variations in the specific surface area of calcined clay in different references, apart from the calcination temperature, is the heating rate that changes temperature to the calcination temperature.

It should be noted that the visual appearance of calcined clay soil obtained through the instantaneous method is quite similar in color to the calcined clay soil obtained through the control method to a considerable extent (Figure 3).

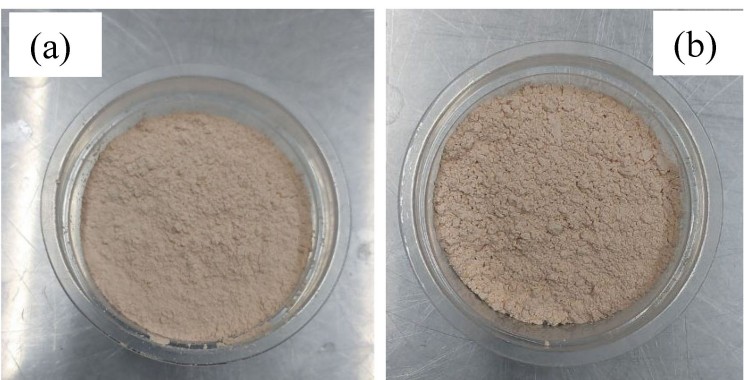

**Figure 3.** Calcinated clay after grinding (**a**) 800 °C/1 H/SL/60 (**b**) 800 °C/1 H/Su/60.

In the study by Fernandez et al. [31], the specific surface area of calcined montmorillonite clay exhibited substantial changes with varying temperatures, while the specific surface area of kaolinite and illite clay types showed minor variations at temperatures between 600 and 800 °C. However, in the research by Ferreiro et al., [32] the specific surface area of calcined kaolinite clay showed noticeable changes at temperatures of 700, 850, and 1000 °C. The findings of the current study regarding the specific surface area show better alignment with the results of Ferreiro et al. [32].

According to ASTM C1240 [40], the pozzolanic activity index of calcined clay was higher than that according to the ASTM C311 [39] method. The reason for this lies in the impact of temperature on the pozzolanic activity of cementitious additives. The pozzolanic activity of these materials increases at higher temperatures. The trend of changes in the pozzolanic activity index is similar to the trend observed for the specific surface area of calcined clay. With an increase in the temperature from 700 to 800 °C, the pozzolanic activity index according to ASTM C311 [39] and ASTM C1240 [40] methods increases by approximately 8% and 5%, respectively. However, with a further increase in temperature to 900 °C, the pozzolanic activity index decreased by around 12% and 6% according to the ASTM C311 [39] and ASTM C1240 [40] methods, respectively. Additionally, the pozzolanic activity index of calcined clay with the control method was approximately 17% and 14% higher than that with the uncontrolled method, according to the ASTM C311 [39] and ASTM C1240 [40] methods, respectively. In the research by Bahman-Zadeh et al. [21], the pozzolanic activity index of calcined clays at 800 °C was assessed according to ASTM C311 [39], which generally corresponded well with the results of the present study.

In the study by Fernandez et al. [29], the effect of calcination temperature on the pozzolanic activity of kaolinite, illite, and montmorillonite clays was investigated using a relatively rapid temperature change method (300 °C/minute) at temperatures of 600 and 800 °C. The results suggest that the pozzolanic activity index of calcined clay is particularly sensitive to the calcination temperature only for the montmorillonite clay type.

Figure 4 presents the X-ray diffraction (XRD) results of calcined clay. The samples are encoded, where the first number denotes the calcination temperature, the subsequent number signifies the retention time (or residence time) at that temperature, and the last character indicates the combustion method. For instance, 800 °C/1 H/SL refers to the sample calcined at 800 °C with a residence time of 1 h and a controlled combustion method. "SL" stands for controlled combustion, and "Su" for immediate combustion.

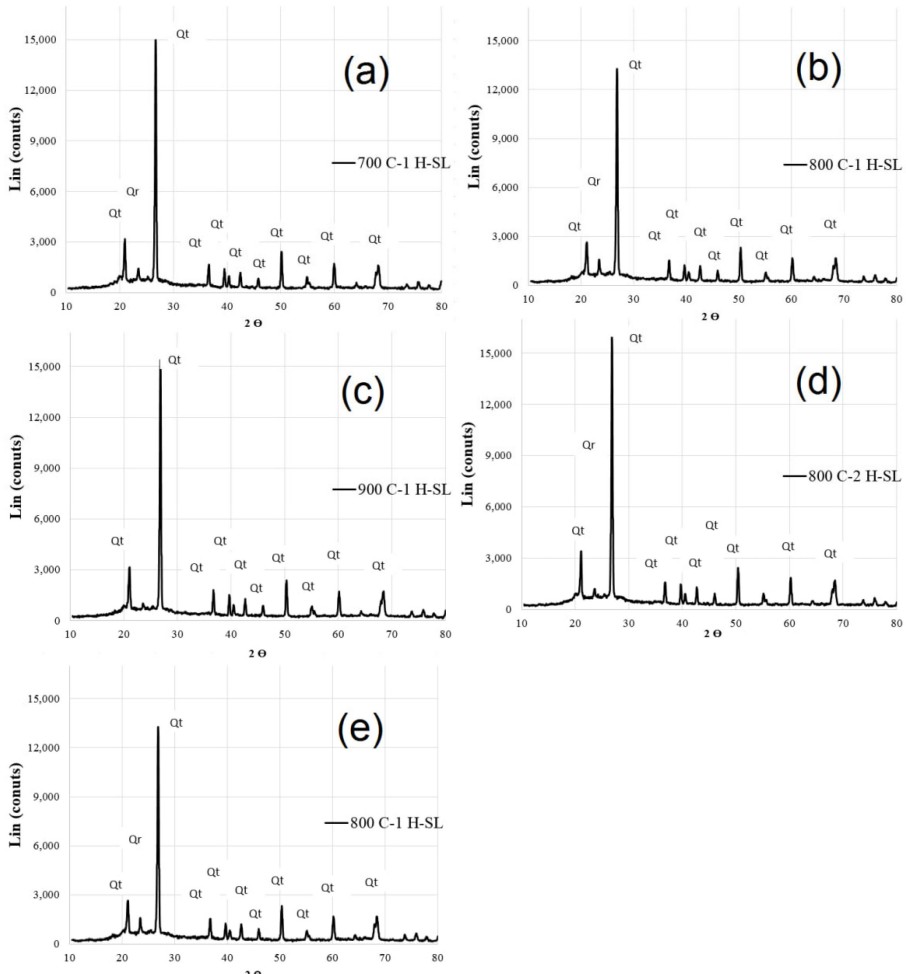

**Figure 4.** XRD Spectra of (**a**) 700 °C/1 H/SL (**b**) 800 °C/1 H/SL (**c**) 900 °C/1 H/SL (**d**) 800 °C/2 H/SL (**e**) 800 °C/1 H/Su. (Qt: Quartz and Qr: Orthoclase).

In the XRD spectrum, the intensity (or counts) of the peaks depends on the number of crystalline constituents at that specific angle. The XRD results, as shown in Figure 3, reveal no trace of kaolinite in the calcined clay, indicating the complete dehydroxylation of kaolinite during the calcination process. The XRD pattern of the calcined clay exhibits peaks corresponding to quartz, feldspar, and mica. In Figure 4a–e, the peak values for different crystals are shown to decrease sequentially. Moreover, the calcined clay demonstrates a high percentage of the amorphous phase, with an 800 °C/1 H/Su sample categorized as low-grade calcined clay due to its high quartz content. In contrast, the calcined clay at 800 °C/1 H/SL displays a high percentage of the amorphous phase, along with the lowest peak in the XRD spectrum (Figure 4b).

It is worth mentioning that a reduction in pozzolanic activity for calcined clay at temperatures exceeding 800 °C is attributed not only to the decrease in the number of amorphous phases due to $SiO_2$ phase changes but also to the alteration in the arrangement of the 5-coordinated Al structure [32,44,46],

### 3.2. Pozzolanic Activity Index

The results of the pozzolanic activity index for different calcined clays are presented in Table 5. It was observed that extending the grinding time beyond one hour (120 min) generally did not have a significant positive effect on the pozzolanic activity index; instead, it resulted in increased water demand. Increasing the grinding time from 15 min up to 60 min led to an increase in both the water demand and the pozzolanic activity index of calcined clay. The pozzolanic activity index of calcined clay mixtures was compared

according to ASTM C1240 [40] and ASTM C311 [39]. While the values of the pozzolanic activity index obtained from the two different methods differed, the impact of temperature and milling duration on the pozzolanic activity index remained consistent in both methods. The variation in the pozzolanic activity index values between these two methods could be attributed to differences in the level of pozzolanic material substitution and variations in processing techniques [47]. Notably, the mixture of calcined clay 800 °C/1 H/SL with a grinding time of 60 min exhibited the best performance compared to other mixtures.

**Table 5.** Pozzolanic activity index of calcined clays.

| Mix Designation | Grinding Time (min) | Accordance to ASTM C311 | | | Accordance to ASTM C1240 |
|---|---|---|---|---|---|
| | | Pozzolanic Activity Index with Portland Cement (%) | | Water Requirement (%) | Accelerated Pozzolanic Strength Activity Indexin (%) |
| | | 7 Days | 28 Days | | |
| 700 °C/1 H/SL | 15 | 60 | 80 | 110 | 115 |
| | 30 | 63 | 82 | 115 | 119 |
| | 60 | 68 | 83 | 120 | 130 |
| | 120 | 65 | 81 | 126 | 130 |
| 800 °C/1 H/SL | 15 | 63 | 84 | 111 | 118 |
| | 30 | 65 | 85 | 116 | 121 |
| | 60 | 69 | 90 | 121 | 136 |
| | 120 | 64 | 84 | 126 | 134 |
| 900 °C/1 H/SL | 15 | 61 | 79 | 112 | 116 |
| | 30 | 63 | 81 | 117 | 117 |
| | 60 | 65 | 81 | 122 | 128 |
| | 120 | 63 | 79 | 130 | 129 |
| 800 °C/2 H/SL | 60 | 66 | 76 | 132 | 120 |
| 800 °C/1 H/Su | 60 | 60 | 75 | 138 | 116 |

The increase in the furnace temperature from 700 °C to 800 °C improved the pozzolanic activity of calcined clay, but further increasing the temperature to 900 °C decreased pozzolanic activity. This decline can be attributed to the crystallization of $SiO_2$ at higher temperatures and the alteration in the arrangement of the 5-coordinated Al structure [31,32]. The enhancement of pozzolanic activity in calcined clay was observed with the elevation of the furnace temperature from 700 °C to 800 °C; however, a subsequent rise to 900 °C led to a decrease in pozzolanic activity. This reduction is ascribed to the crystallization of $SiO_2$ at elevated temperatures and a modification in the arrangement of the 5-coordinated Al structure [31,32,46]. Additionally, the specific surface area of calcined clay at 900 °C was lower than that at 800 °C, potentially leading to a reduced pozzolanic activity index.

Increasing the grinding time up to 60 min in calcined clay at a specific temperature enhanced the pozzolanic activity index and extending the grinding time to 120 min resulted in an excessive reduction in particle size and irregular particle shapes. As a consequence, calcined clay particles accumulated in the cement mixture, leading to a decrease in the pozzolanic activity index.

Comparing the pozzolanic activity index of calcined clay at 800 °C/1 H/Su with a grinding time of 60 min to that of calcined clay at 800 °C/1 H/SL with the same grinding time, it was evident that immediate combustion reduced the quality of calcined clay. This reduction in pozzolanic activity in flash-calcined clay could be attributed to a decrease in the uniformity of the formed structure [48].

Furthermore, when comparing the pozzolanic activity index of calcined clay at 800 °C/1 H/SL to that of calcined clay at 800 °C/2 H/SL, it indicated that increasing the calcination time at a specific temperature could reduce the pozzolanic activity of calcined

clay. This finding underscores the importance of determining the optimum calcination time at the desired temperature to achieve calcined clay with superior pozzolanic performance.

### 3.3. Workability of Concretes

The workability of concrete mixes can be assessed by examining the percentage of superplasticizers in cement required to maintain workability at the level of the control mix. Table 6 presents the amount of superplasticizer required for each mix in comparison to the control mix. The results indicate that the superplasticizer content increased with higher calcination temperatures. Additionally, increasing the grinding time also led to an increase in the superplasticizer requirement for concrete.

**Table 6.** Superplasticizer content of mixtures.

| Mix Designation | SP (% Cement) |
|---|---|
| Ref | 0.47 |
| 700 °C/1 H/SL/15 | 0.67 |
| 700 °C/1 H/SL/30 | 0.71 |
| 700 °C/1 H/SL/60 | 0.76 |
| 700 °C/1 H/SL/120 | 0.83 |
| 800 °C/1 H/SL/15 | 0.72 |
| 800 °C/1 H/SL/30 | 0.75 |
| 800 °C/1 H/SL/60 | 0.79 |
| 800 °C/1 H/SL/120 | 0.87 |
| 900 °C/1 H-SL/15 | 0.73 |
| 900 °C/1 H/SL/30 | 0.78 |
| 900 °C/1 H/SL/60 | 0.82 |
| 900 °C/1 H/SL/120 | 0.94 |
| 800 °C/2 H/SL/60 | 0.73 |
| 800 °C/1 H/Su/60 | 0.72 |

Notably, the results in the table show that the amount of superplasticizer used in the 800 °C/2 H/SL/60 and 800 °C/1 H/Su/60 mixtures was lower than that in the 800 °C/1 H/SL/60 mixture. This observation could be attributed to the lower specific surface area of the calcined clay mixtures, at 800 °C//2 H/SL/60 and 800 °C/1 H/Su/60, in comparison to the calcined clay mixture at 800 °C/1 H/SL/60.

### 3.4. Compressive Strength

Figure 5 shows compressive strength values at different ages for up to 180 days. The compressive strength results show that changing the calcination method could significantly affect the compressive strength of the concrete. The results show that the compressive strength increased when the calcination temperature increased from 700 °C to 800 °C. However, when the temperature increased to 900 °C, the compressive strength decreased slightly. The reason for this reduction in compressive strength is the increase in the crystalline phases of calcined clay (as indicated by the XRD results (Figure 3)). The results of other researchers also show an increase in crystalline phases at temperatures over 800 °C [32,44,46–48].

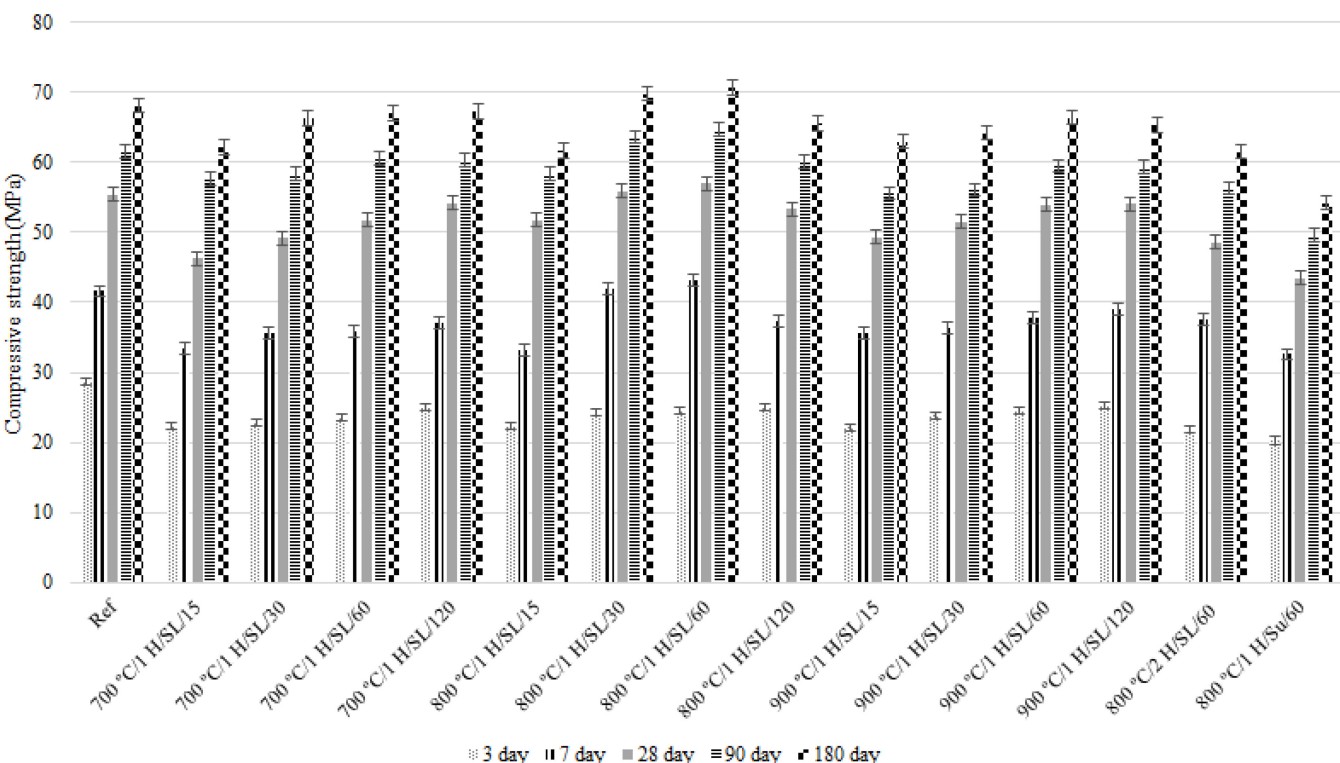

**Figure 5.** The compressive strength of mixtures.

An increase in compressive strength was observed as the grinding time of the calcined clay increased. With the optimum processing method for clay calcination and suitable clay, it is possible to achieve the same strength in LC3 concrete as in Portland cement concrete.

The results show that the compressive strength of 800 °C/1 H/SL/60 and 800 °C/2 H/SL/30 for concrete are higher than the compressive strength of the control concrete. The optimum grinding time of the calcined clay for this type of raw clay, calcined at 800 °C in a controlled combustion method, took 1 h to achieve the maximum compressive strength of LC3 (about 70.9 MPa in 180 days).

The 28-day compressive strength of the mixture at 800 °C/2 H/SL/60 decreased by about 15% compared to the mixture at 800 °C/1 H/SL/60. This result shows that the duration of the clay in the kiln for the calcination of clay at a certain temperature exhibited an optimum value. For the type of soil studied here, a duration of more than one hour at 800 °C decreased the quality of the calcined clay produced and, consequently, the compressive strength.

The 28-day compressive strength of the mixture at 800 °C/1 H/SL/60 was about 24% higher than the compressive strength of the mixture at 800 °C/1 H/Su/60.

The calcined clay at 800 °C exhibited the highest glassy phase content and a more suitable 5-coordinated Al structure [32,44,46]. Additionally, due to the fine particle size of calcined clay, it can be interspersed among coarser particles and provide a higher density in the mixture. As a result of the rapid pozzolanic reaction in the calcined clay and the appropriate density of the mixture, the compressive strength of the 800 °C/1 H/SL/60 mixture increased compared to the control mixture. It is worth noting that calcined clay processed at 800 °C with a milling duration of 60 min showed the best pozzolanic performance among the clays prepared in this study.

*3.5. Rapid Chloride Migration Test*

Figures 6 and 7 show the chloride diffusion coefficients (which were measured with RCMT methods) of mixtures at different ages from 3 to 180 days. The results show how chloride diffusion coefficients decreased with an increase in the age of concrete. The trend

of RCMT test results of LC3 mixtures is different from the trend of compressive strength test results. The pozzolanic reaction of calcined clay and limestone powder increased the resistance of concrete against the penetration of chloride ions and, as a result, reduced the diffusion coefficient of concrete. The chloride diffusion coefficients of LC3 mixtures at 90 and 180 days were lower than the chloride diffusion coefficients of the Ref. mixture.

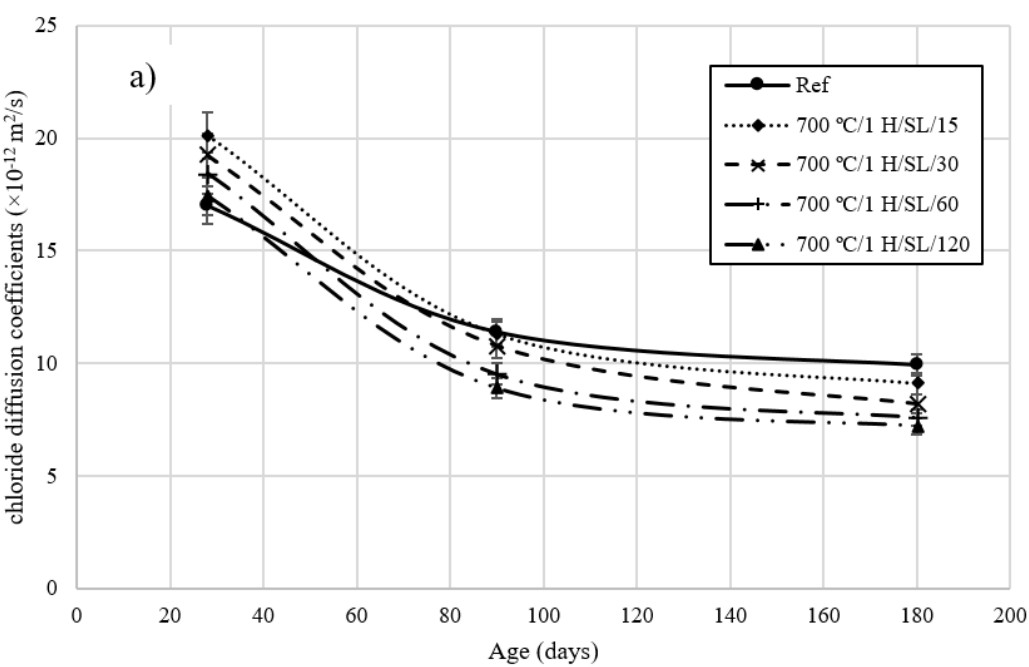

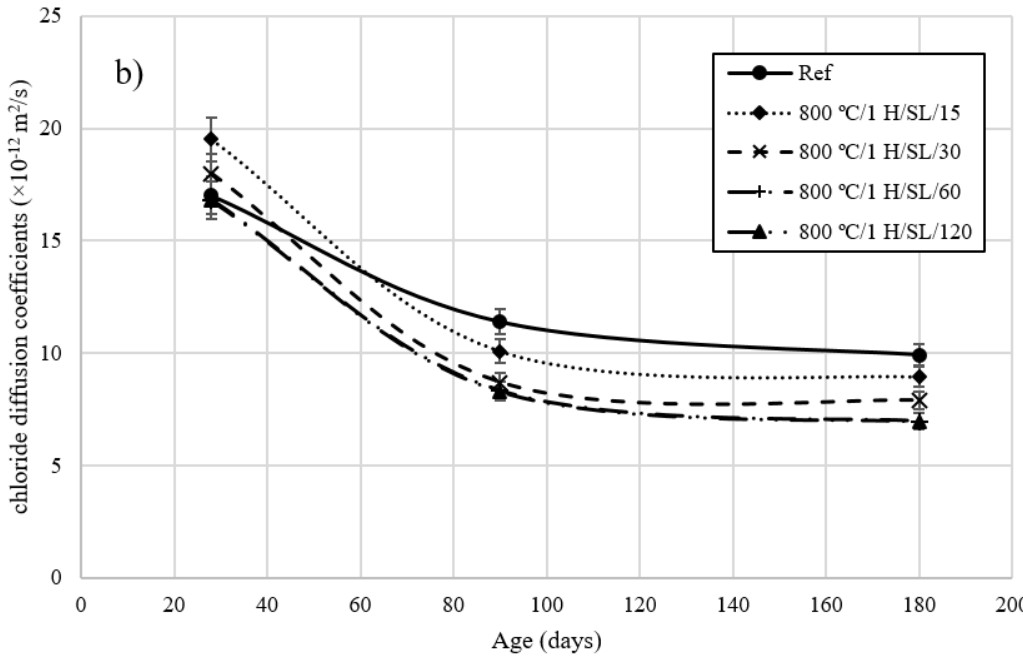

**Figure 6.** *Cont.*



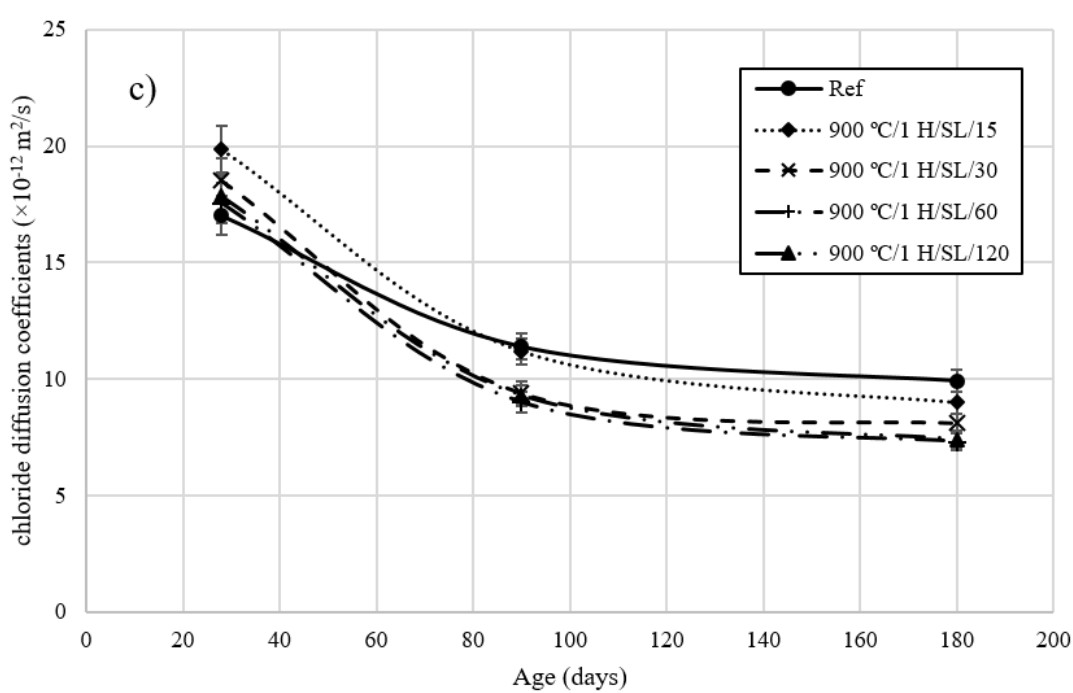

**Figure 6.** Effect of temperature and time grinding on chloride diffusion coefficients (**a**) Mixture with calcined clay at 700 °C/1 H/SL; (**b**) Mixture with calcined clay at 800 °C/1 H/SL; (**c**) Mixture with calcined clay at 900 °C/1 H/SL.

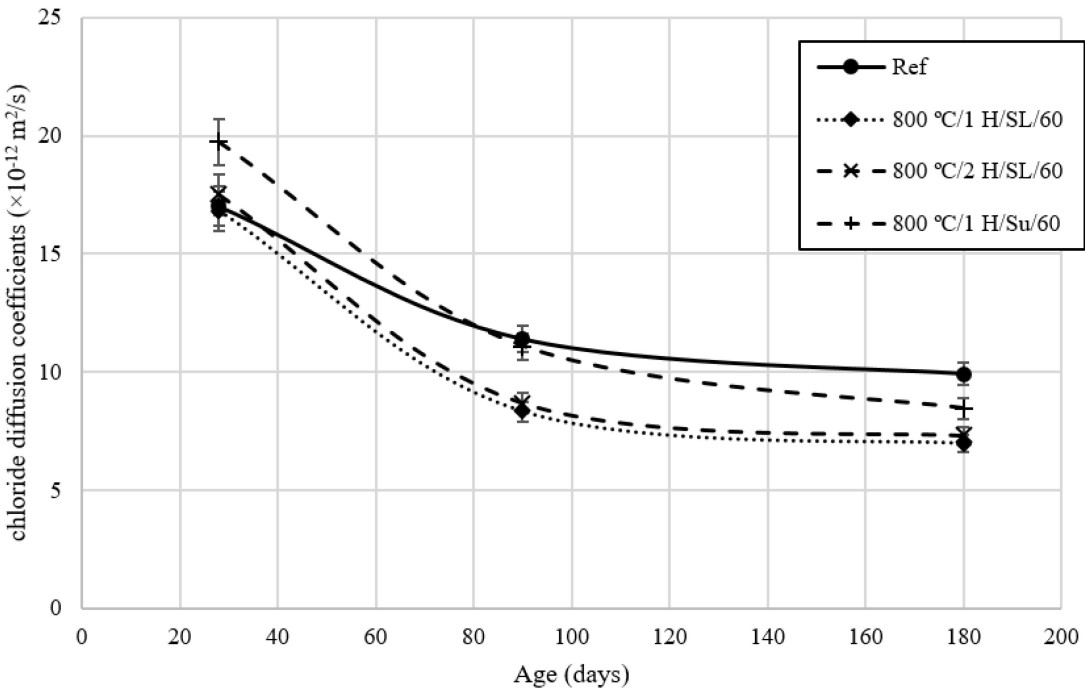

**Figure 7.** Effect of combustion methods and duration of calcination on chloride diffusion coefficients.

The chloride diffusion coefficient results show that changing the calcination method significantly affected the chloride diffusion coefficients of the concrete. These results show that chloride diffusion coefficients decreased when the calcination temperature increased from 700 °C to 800 °C. When the temperature was increased from 800 °C to 900 °C, the chloride diffusion coefficients decreased slightly. A decrease in chloride diffusion coefficients was observed as the grinding time of calcinated clay increased.

The 180-day chloride diffusion coefficients of the mixture 800 °C/1 H/SL/120 were about 24% lower than the compressive strength of the Ref. mixture. The results show that the optimum grinding time of calcined clay for this type of raw clay, calcined at 800 °C in a controlled combustion method, took 2 h to achieve better performance compared to the chloride penetration resistance of the LC3 mixture, which this research has investigated.

## 4. Prediction of Corrosion Initiation

Numerical modeling methods were employed to predict the time of reinforcement corrosion initiation in concrete. Fick's second law of diffusion was utilized to describe the penetration of chloride ions into the concrete. Assuming a constant surface chloride ion concentration and constant chloride diffusion coefficient, the differential equation of Fick's second law was solved using the Crank–Nikolson numerical method based on finite difference, as shown in Equation (2) [49,50]:

$$C(x,t) = C_i + (C_s - C_i)\left[1 - erf\left(\frac{x}{2\sqrt{D_a\,t}}\right)\right] \tag{2}$$

where $C(x,t)$ is the chloride concentration (% mass of concrete) at the depth $x$ and the time $t$, $C_i$ is the initial chloride concentration (% mass of concrete) before exposure to the chloride environment, $C_s$ represents the surface chloride concentrations (% mass of concrete) on the exposed surface, $erf$ is the error function, $x$ is the depth below the exposed surface $(m)$, $t$ is the exposure time (s), and $D_a$ is the apparent diffusion coefficient of chlorides $\frac{m^2}{s}$.

To compare the concrete's resistance against chloride penetration, the Crank–Nikolson method was applied based on the results of Rapid Chloride Migration (RCM) tests. The changes in chloride diffusion coefficient over time were determined using Equation (3) [49,50]:

$$D(t) = D_{ref}\left(\frac{t_{ref}}{t}\right)^m \tag{3}$$

where $D(t)$ is the concrete diffusion coefficient at time $t$, $D_{ref}$ is the concrete diffusion coefficient at time $t_{ref}$, and $D$ is the aging factor.

In Table 7, the aging factor m was calculated based on Equation (3), and the 28-day chloride coefficient diffusion obtained from the RCM method was presented for use in the modeling.

**Table 7.** Age factors and the reference diffusion coefficients.

| Mix ID | $D_0$ ($\times 10^{-12}$ m²/s) | m |
|---|---|---|
| Ref | 17.02 | 0.295 |
| 700 °C/1 H/SL/15 | 20.12 | 0.433 |
| 700 °C/1 H/SL/30 | 19.24 | 0.461 |
| 700 °C/1 H/SL/60 | 18.43 | 0.485 |
| 700 °C/1 H/SL/120 | 17.46 | 0.485 |
| 800 °C/1 H/SL/15 | 19.51 | 0.434 |
| 800 °C/1 H/SL/30 | 17.98 | 0.461 |
| 800 °C/1 H/SL/60 | 16.82 | 0.486 |
| 800 °C/1 H/SL/120 | 16.79 | 0.486 |
| 900 °C/1 H/SL/15 | 19.87 | 0.433 |
| 900 °C/1 H/SL/30 | 18.53 | 0.458 |
| 900 °C/1 H/SL/60 | 17.55 | 0.481 |
| 900 °C/1 H/SL/120 | 17.84 | 0.481 |
| 800 °C/2 H/SL/60 | 17.51 | 0.482 |
| 800 °C/1 H/Su/60 | 19.74 | 0.459 |

The critical chloride concentration in this study was considered to be equal to 0.18% of the concrete weight, and the surface chloride concentration was fixed at 0.8% of the concrete weight [49,50]. It should be noted that, for submerged conditions in the Life 365 software, the surface chloride concentration is fixed at 0.8 for certain conditions [51].

Figures 8 and 9 present the predicted time of corrosion initiation at different depths for the studied mixtures, assuming a constant surface chloride concentration and a constant chloride diffusion coefficient after 25 years. The results demonstrate that the time of corrosion initiation for LC3 mixtures was significantly higher than that of the Ref mix for reinforcement covers above 5 cm. Figure 10 displays the corrosion initiation time of the mixtures at a depth of 50 mm. The corrosion initiation time for all LC3 mixtures investigated in this research was longer than the control mixture due to the pozzolanic reaction of calcined clay.

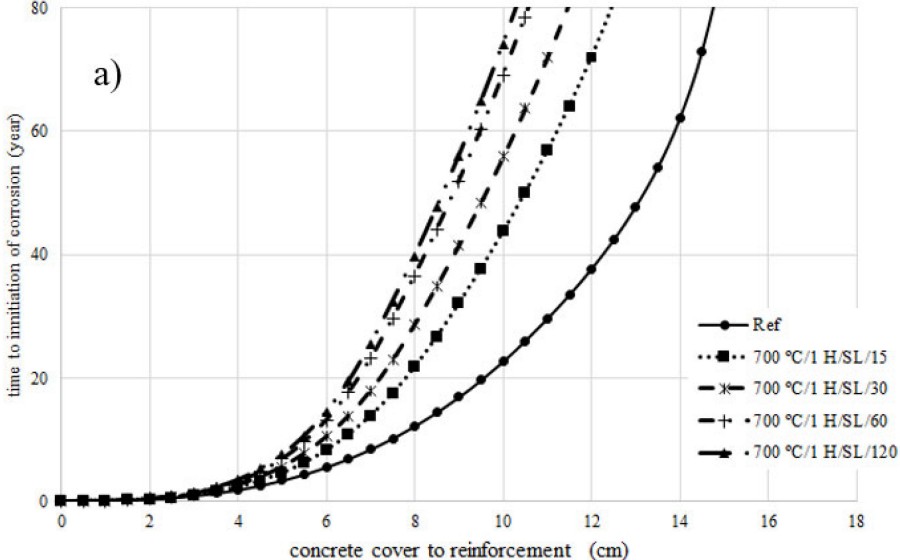

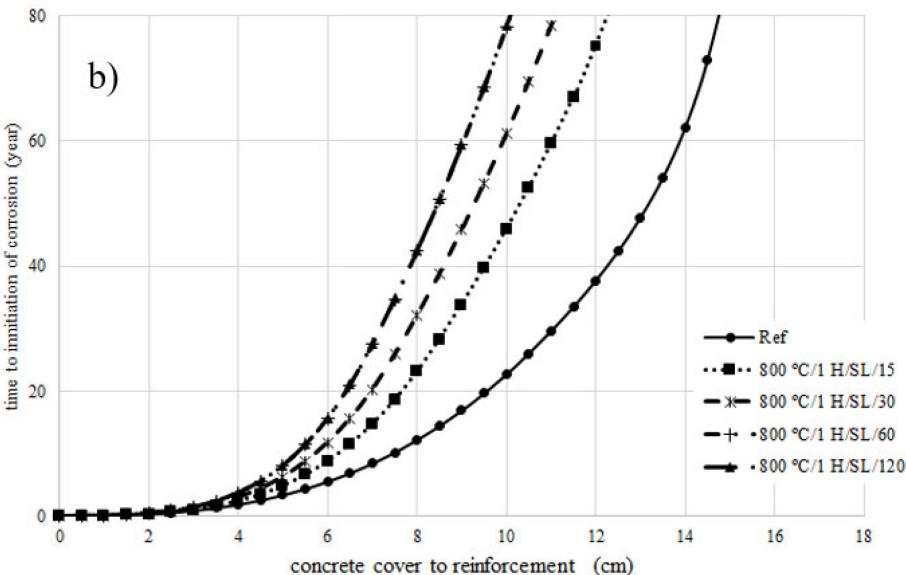

**Figure 8.** *Cont.*

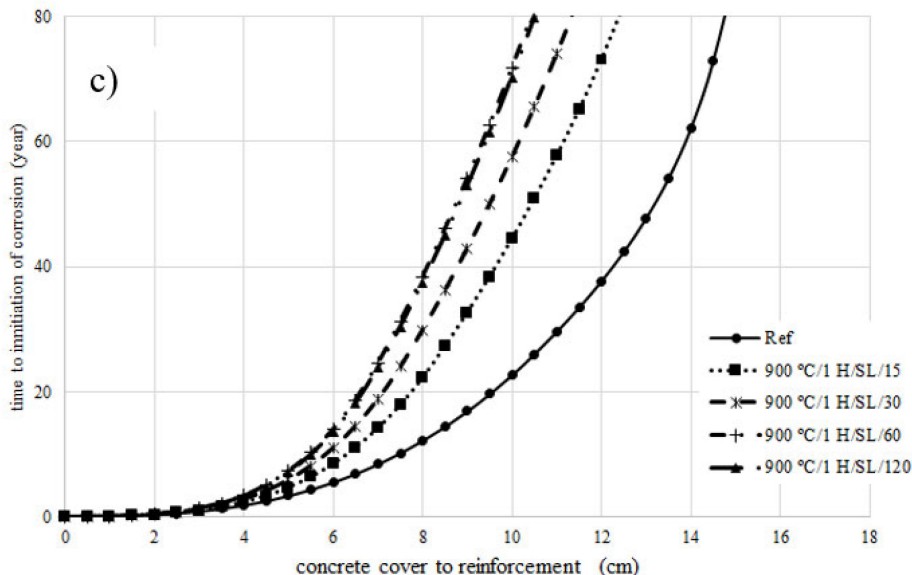

**Figure 8.** Predicted time of corrosion initiation at different depths for (**a**) Mixture with calcined clay at 700 °C/1 H/SL, (**b**) Mixture with calcined clay at 800 °C/1 H/SL, and (**c**) Mixture with calcined clay at 900 °C/1 H/SL.

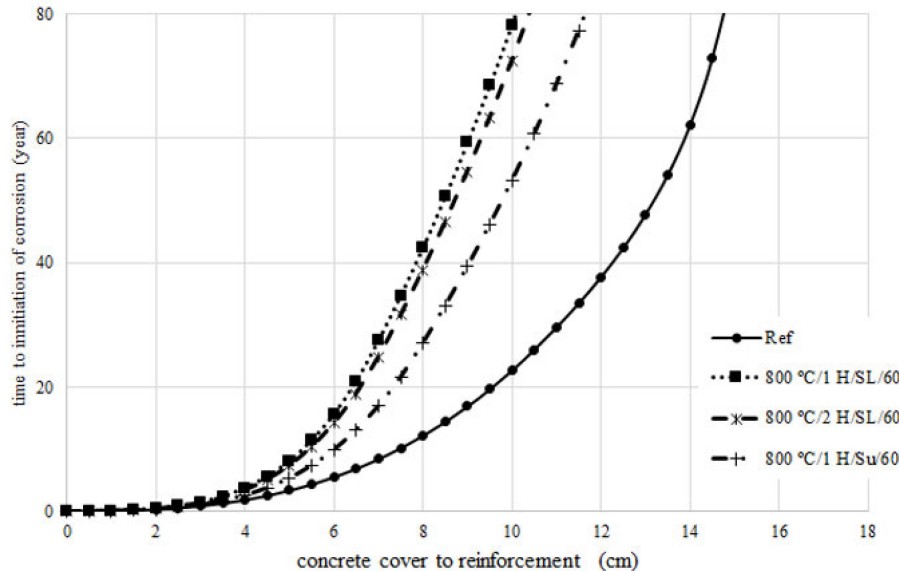

**Figure 9.** Effect of combustion methods and duration of calcination on predicted time of corrosion initiation at different depths.

Among the investigated calcination temperatures of raw clay, an increase in the grinding time of calcined clay led to a significant increase in the time of reinforcement corrosion. The mixture 800 °C/1 H/SL/120 showed the longest corrosion initiation time among the investigated mixtures (about 20% longer compared to the Ref mixture).

At a constant grinding time, increasing the calcination temperature from 700 °C to 800 °C resulted in an increased corrosion initiation time. However, with a further increase in temperature from 800 °C to 900 °C, the corrosion initiation time decreased.

Comparing the corrosion initiation time of mixtures 800 °C/2 H/SL/60 and 800 °C/1 H/SL/60, it is evident that increasing the calcination time from one hour to two hours reduced the corrosion initiation time by about 9%. This indicates the negative effect of increasing the calcination time on the performance of calcined clay in concrete.

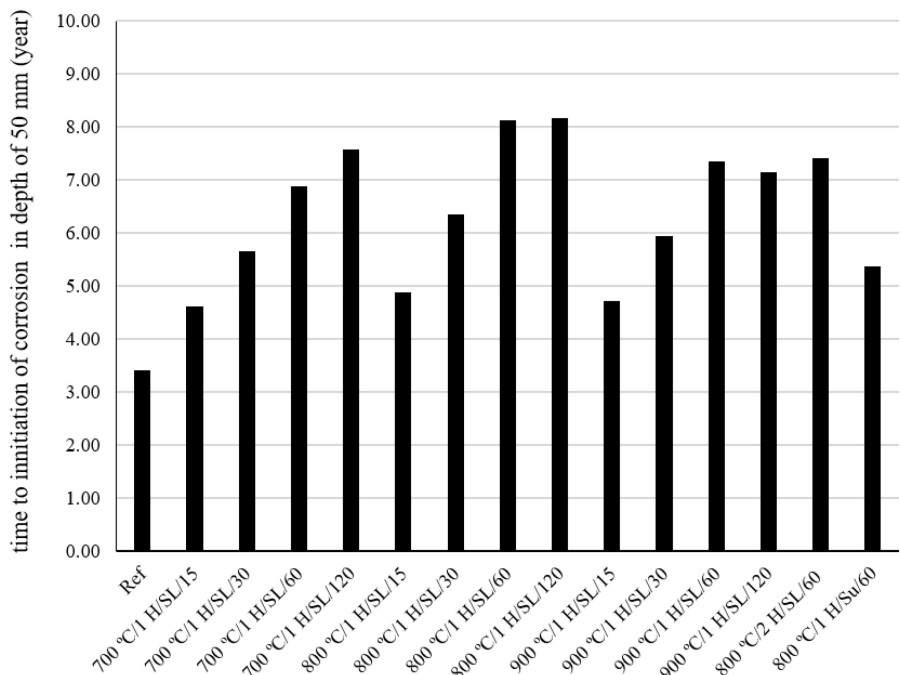

**Figure 10.** Predicted time of the corrosion initiation at depths of 50 mm.

Furthermore, when comparing the corrosion initiation time of the mixtures 800 °C/1 H/SL/60 and 800 °C/1 H/Su/60 at a 5 cm cover depth, it was found that the mixture with clay calcined using the controlled combustion method had an approximately 34% higher corrosion initiation time compared to the mixture with clay calcined using the instantaneous combustion method.

Interestingly, the mixture with the highest corrosion initiation time did not necessarily have the highest compressive strength.

Among the investigated mixtures, the mixture composed of calcined clay that was processed at 800 °C and milled for 60 min exhibited the highest compressive strength at ages ranging from 7 to 180 days. Furthermore, the onset of corrosion for the mixture made with this calcined clay was significantly delayed compared to both the control mixture and the mixture containing other calcined clays. It is noteworthy that the corrosion onset time for the 800 °C/1 H/SL/60 mixture and the 800 °C/1 H/SL/120 mixture was almost the same. However, the compressive strength of the 800 °C/1 H/SL/60 mixture was higher than that of the 800 °C/1 H/SL/120 mixture. Based on the obtained results, the optimal temperature for clay calcination is around 800 °C, and the ideal milling duration is 60 min. Furthermore, the results of this study indicate that higher calcination temperatures and longer milling durations do not necessarily result in clays with better performance in cementitious mixtures.

These findings can be utilized to optimize the costs associated with grinding and combustion temperature in the industrial production process of calcined clay.

## 5. Conclusions

Based on the findings of this study on the LC3 and control mixes, the following conclusions can be drawn:

1.  By increasing the temperature from 800 °C to 900 °C, the pozzolanic index according to ASTM C1240 and ASTM C311 over 28 days decreased by approximately 4% and 7%, respectively. This indicates that at temperatures above 800 °C, the pozzolanic activity of calcined clay diminishes. The pozzolanic index of the calcined clay at 800 °C using the controlled combustion method, as per ASTM C1240 and ASTM C311 in 28 days, was 17% and 20% higher than the pozzolanic index of calcined clay at 800 °C using the

instantaneous combustion method. This suggests that the instantaneous combustion method has led to a reduction in the pozzolanic activity of calcined clay.

2. The pozzolanic performance of calcined clay in the presence of a superplasticizer is slightly better than its performance in mixtures without a superplasticizer.

3. The 800 °C/1 H/SL/60 mixture exhibits the highest compressive strength among all investigated LC3 mixtures and the control mix. Its compressive strength development process is also desirable. Additionally, the 28-day compressive strength experienced by the 800 °C/1 H/SL/30 mixture was higher than that of the control mix and comparable to the compressive strength development of the 800 °C/1 H/SL/60 mixture.

4. The chloride diffusion coefficient of the 800 °C/1 H/SL/120 mixture was lower than that of the other investigated LC3 mixtures. Moreover, the time to corrosion initiation for the 800 °C/1 H/SL/120 mixture was higher than that of the other investigated mixtures.

5. Based on these results, the optimal temperature for clay calcination is about 800 °C, and the optimal grinding time is about 60 min.

6. The compressive strength and the onset of corrosion initiation time for the LC3 mixture containing calcined clay produced by the instantaneous calcination method, in comparison to the LC3 mixture containing calcined clay produced by the controlled calcination method, experienced a reduction of 24% and 34%, respectively. The instantaneous combustion of clay reduced the quality of calcined clay.

These findings underscore the significance of factors such as the heating rate, calcination temperature, and grinding duration when optimizing the effectiveness of calcined clay as an additive in LC3 concrete. These insights make valuable contributions to the advancement of eco-friendly concrete blends, which not only lower $CO_2$ emissions but also enhance their durability.

**Author Contributions:** Conceptualization and Methodology, Supervision, Review: R.M. and M.M.R.; Investigation, Methodology, Data Curation, Formal Analysis, Writing—Original Draft, Writing & Editing: S.N. All authors have read and agreed to the published version of the manuscript.

**Funding:** This research received no external funding. The APC was funded independently.

**Data Availability Statement:** All data supporting the findings of this study are available within the article.

**Conflicts of Interest:** The authors declare no conflict of interest.

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
