# Peer review of "Investigating the Calcination Temperature and Grinding Time of Calcined Clay on the Mechanical Properties and Durability of LC3 Concrete"

_infrastructures, doi:10.3390/infrastructures8100139_

Round 1
Reviewer 1 Report
The manuscript entitled “Investigation of calcination temperature and grinding time of calcined clay on mechanical properties and durability of LC3 concrete” studied the impact of calcination temperature and grinding time on mechanical properties and durability of limestone calcined clay concrete. In general, the manuscript should be improved before acceptance for publication. It is recommended that the authors also discuss their results regarding granulometric measurements and pay attention to the explanation regarding SiO2 crystallization since the temperatures are too low.
· Authors must inform the origin and type of clay and limestone used in this work.
· The manuscript, there are many spelling errors.
· When citing an abbreviation for the first time in the manuscript, please provide the description.
· What means “retention time”? Isothermal treatment?
· Authors must remove all experimental results shown in the “experimental program” section to “Results and discussions.”
· The paragraph between 78-79 lines is a little confusing. In the beginning, the authors explain that the clays were calcinated at 700 ºC, 800 ºC, and 900 ºC 72 for a duration of 1 hour. Still, at the end of the paragraph, the authors wrote, “For this investigation, the calcined clay was produced at a temperature of 800 ºC 78 and a grinding time of 60 minutes”. How was the correct information?
· The authors provide more information about the “controlled combustion method” and “immediate combustion,” which looks like the atmosphere; the materials were added to some crucible and equipment where such treatments were performed.
· Please provide information about the equipment (brand and model) on which the specific surface area measurements were taken. Do the same for the other experiments.
· At line 121, correct “Testsperformed” to “Tests performed”
· Maybe the information in a sentence in lines 153-154, “It was observed that increasing the grinding time beyond one hour generally had no significant positive effect on the pozzolanic activity index,” is contradictory with the information in a sentence in lines 155-156 “Longer grinding times 154 led to increased water demand and pozzolanic activity index of the calcined clay”.
· About the sentence in lines 155-158: “Comparing the pozzolanic activity index of calcined clay mixtures according to ASTM C1240 156 [32] and ASTM C311 [31], it was found that the former method yielded slightly better 157 results, but overall, the results were similar”, I believe that the values is not similar but follows the same tendency, please confirm!
· At lines 161-166, the authors conclude that pozzolanic activity does not increase with increasing temperature from 800°C to 900° is credited to SiO2 crystallization. Are the authors sure about this? This temperature is too low for the crystallization of SiO2, and at most, polymorphic transformations may occur. Also, are the authors sure that reference [25] explains their results regarding SiO2 crystallization?
· To substantiate their results in terms of granulometry, it is recommended that authors add granulometry measurements.
· The figura 3 is missing.
· The subsection “3.4. Rapid chloride migration test” discussion is entirely wrong.
· At line 283, what does mmm?
· The authors should support conclusion 1 with experimental results.
· What does superplasticizer mean in the manuscript context? Were they added to the mix?
· At lines 362-364, in the sentence: “Moreover, the time to corrosion initiation of the 800 ºC-1 H-SL-120 mixture is better than that of the other investigated mixtures.”, what means “better”?
Author Response
Dear Reviewer,
We deeply appreciate your thoughtful review of our manuscript titled "Investigation of calcination temperature and grinding time of calcined clay on mechanical properties and durability of LC3 concrete." Your feedback has been immensely valuable in improving the quality of our research.
In response to your insightful recommendations, we have diligently revised the manuscript, making significant changes to its content. These revisions address the points you raised and aim to enhance the overall clarity and quality of our paper.
We are pleased to inform you that the revised manuscript, along with the file containing the implemented revisions, has been included as attachments to this email.
Thank you once again for your valuable input and constructive feedback. Your expertise has played a pivotal role in refining our work.
Please see the attachment
Sincerely,
Sina Nasiri

Reviewer 2 Report
Please see in the attachment.

.
Author Response

(The authors gave the same response as above.)

Reviewer 3 Report
This manuscript investigates the effect of calcination temperature and grinding time of clay on the mechanical properties and durability of LC3 concrete. The topic is of interest, and the experimental design is reasonable. However, the empirical data and relevant analyses are undeveloped which do not meet discipline standards. For example, the authors should add more discussion on the effect mechanism instead of simple data enumeration. Thus, the reviewer would suggests a major revision for this manuscript. For details, please see below:
In table 1, could the authors explain why calcined clay (900) has a higher LOI than calcined clay (800)?
Please indicate the meaning of Qt in Figure 1. Is it quartz? Also, a phase named Qr appears in Figure 1. What is it?
The reviewer believes the XRD pattern of raw clay should be included in Figure 1. This can give crucial information about clay minerals in the raw material – kaolinite, illite, montmorillonite, or others?
Tables 2 and 4 – please use a consistent font format. In the current version, some texts are bolded while others are not.
Lines 163-164 “This decline is attributed to the crystallization of SiO2 at higher temperatures [25], as confirmed by the XRD test results”
The reviewer checked the reference [25] but didn’t find the statement about SiO2 crystallization at 900 ℃. Also, the authors said the crystallization of SiO2 can be confirmed by changes in XRD patterns, but the reviewer did not see a significant change in Figure 1.
In fact, a previous study has confirmed that the pozzolanic activity of clay minerals is closely related to the appearance of 5-coordinated Al, see doi.org/10.1016/j.cemconres.2010.09.013. In this study, the highest firing temperature is 900 ℃, at which SiO2 usually hardly crystallize. Even the formation of spinel/mullite in metakaolin at ~980℃ appears in Al-rich grains instead of Si-rich regions.
From the reviewer’s point of view, the decreased pozzolanic activity from 800 to 900 ℃ is caused by 1) the improved Si/Al-O order or crystallographic order as dehydroxylated clay minerals prepare themselves for crystallization with increased temperature and 2) the sintering of dehydroxylated clay minerals. For kaolinite, illite and montmorillonite, a sharp volumetric reduction usually occurs at ~900 ℃, see doi.org/10.1016/j.jobe.2022.105802; thus the particle size and specific surface area are significantly changed.
The reviewer thus invites authors to re-edit this sentence and replace reference [25] with the above articles.
Lines 202-203 “The reason for this reduction in compressive strength is the increase of crystalline phases in calcined clay (as indicated in the XRD results (figure 1)).”
Again, the reviewer has not seen increased crystal phases from 800 to 900℃ in Figure 1. Please argue or modify the description here.
Section 3.4 In this section authors hit the new level of careless known by the reviewer. Section 3.4 entitled “Rapid chloride migration test” but delivers nearly the same content as “Section 3.3. Compressive strength”. Authors must supply new data analysis on rapid chloride migration test before this manuscript could be possibly reviewed.
There are four curves but five legends in Figure 7b or 7c. Authors must provide missing curves here.
Lines 316-317 “This result suggests that the difference in performance between calcined clay with instantaneous and controlled combustion methods is not significant.”
Please note that 34% is a pretty significant change.
Line 323 “the optimum temperature range for calcination lies between 700 ºC and 800 ºC”
How did the authors draw this conclusion? Please detail it in the manuscript.
Author Response

(The authors gave the same response as above.)

Reviewer 4 Report
This experimental study accounts the effect of temperature and grinding time on the mechanical behavior of LC3. The authors are expected to address the below given comments, provide reasoning and make references to the various ideas brought about to improve the quality and readability of this manuscript before it can be considered for revaluation.
1. The authors should clarify the novelty of the study more explicitly.
2. Can you elaborate on the mechanisms through which the calcination temperature affects the glass phase content and the pozzolanic activity index of the calcined clay, particularly in comparison to the controlled combustion and instantaneous combustion methods?
3. Explain the rationale behind the optimal calcination temperature range of 700°C to 800°C and the grinding time of approximately 60 minutes. How do these conditions synergistically affect the mineralogical and microstructural properties of the calcined clay, resulting in improved performance in LC3 concrete?
4. Given the varying calcination temperatures and grinding times, can you elucidate the potential influence of particle size distribution and surface area of the calcined clay on the microstructure development and hydration kinetics of LC3 concrete?
5. Introduction: Line: 28-28, Provide the following reference, https://doi.org/10.1016/j.jclepro.2022.133492.
6. Could you provide an in-depth analysis of the potential chemical reactions and phase transformations occurring within the calcined clay particles at varying temperatures and grinding times, focusing on their role in influencing the mechanical and durability aspects of LC3 concrete?
7. There are many grammatical mistakes. Pay attention to it!
Moderate editing of English language required.
Author Response

(The authors gave the same response as above.)

Round 2
Reviewer 1 Report
The manuscript was updated by the authors, however, it still need corrections before publication. See below:
· Line 140: is not necessary to insert 800 degree of celsius (°C), just 800 °C.
· Retention time = isothermal treatment; please check it!
· Table 4: The authors state that in Table 1 is possible to see “bulk density, residue on a 45-micron sieve, specific surface area using the Blaine and nitrogen adsorption methods, and the pozzolanic activity of the calcined clays and stone powder are presented” but some theses results is not there, please check it!
· Lines 290-291: Please show the reader where on the graphs they can confirm the following sentence: “Conversely, the calcined clay of 800 °C/1 H/SL displayed a high percentage of the amorphous phase.”
· Between lines 324-326: the authors wrote the following information “It was observed that increasing the grinding time beyond one hour generally had no significant positive effect on the pozzolanic activity index; however, it led to increased water demand.” BUT between lines 326-327, the authors wrote a contradictory sentence “Longer grinding times led to increased water demand and pozzolanic activity index of the calcined clay.” Which one is correct? The activity index of the calcined clay increases or decrease with longer grinding times?
· Please provide a reference for the following sentence “The variation in the pozzolanic activity index values between the two methods can be attributed to differences in the level of pozzolanic material substitution and variations in the processing techniques.”.
· The authors must transcribe in the response to the reviewer where exactly in references [27][28][42] it is possible to draw the conclusion of the sentence: “The formation of crystalline quartz phases could be the reason for the decrease in specific surface area at 900 °C compared to 800 °C [27][28][42].”
· The Authors must add photos to support what was said in the sentence contained between lines 248-250: “It should be noted that the visual appearance of the calcined clay soil obtained through the instantaneous method is quite similar in color to the calcined clay soil obtained through the controlled method, to a considerable extent”.
· At lines 292-293: please explain this sentence “In Figure 3, the higher peaks at each angle correspond to a greater number of crystals associated with that mineral.”
· Line 296: change “800 degrees”.
· Please transcribe for the reviewer which part of references [27][42] supports the sentence: “It is worth mentioning that the reduction in the pozzolanic activity of the calcined clay at temperatures exceeding 800 degrees is attributed not only to the decrease in the amount of amorphous phases due to SiO2 phase changes but also to the alteration in the arrangement of the 5-coordinated Al structure”.
· Please transcribe for the reviewer which part of references [28][27] supports the sentence: “This reduction is ascribed to the crystallization of SiO2 at elevated temperatures and a modification in the arrangement of the 5-coordinated Al structure”.
Author Response
Dear Reviewer,
I hope this message finds you well. I wanted to express my sincere gratitude for your thoughtful review of our manuscript titled "Investigation of calcination temperature and grinding time of calcined clay on mechanical properties and durability of LC3 concrete." I'm writing to inform you that we have taken your insightful recommendations to heart and have diligently revised the manuscript.
I am pleased to attach the revised manuscript, along with the file containing all implemented revisions, to this email. Your expertise and constructive feedback have played a crucial role in refining our work, and we are grateful for your efforts in ensuring the excellence of our research.
Best regards,
Sina Nasiri

Reviewer 2 Report
Accept in present form.
Minor editing of English language required.
Author Response
Dear Reviewer,
I hope this email finds you well. I wanted to take a moment to express my gratitude for your time and effort in reviewing our manuscript, titled "Investigation of calcination temperature and grinding time of calcined clay on mechanical properties and durability of LC3 concrete."
The feedback and insights we receive from reviewers like you are essential for the improvement and quality assurance of our work. Your dedication to the peer-review process is commendable, and we appreciate your role in ensuring the scholarly integrity of our publication.
Once again, thank you for your time and consideration.
Best regards,
Sina Nasiri
Reviewer 3 Report
Authors have revised this manuscript according to the reviewer’s comments, so the manuscript can be accepted for publication now.
Author Response

(The authors gave the same response as above.)

Reviewer 4 Report
The authors have addressed my comments, the manuscript is acceptable for publication in its current form.
Minor editing of English language required
Author Response

(The authors gave the same response as above.)

Round 3
Reviewer 1 Report
accepted!